# Assessment of the Impact of COVID-19 Lockdown on the Nutritional Status and Lipid Profile of Employees in a Teaching Hospital in Rome: A Retrospective Cohort Study

**DOI:** 10.3390/ijerph19084549

**Published:** 2022-04-09

**Authors:** Lorenza Lia, Eleonora Ricci, Corrado Colaprico, Eleonora Di Legge, Augusto Faticoni, Lorenzo Maria Donini, Giuseppe La Torre

**Affiliations:** 1Department of Public Health and Infectious Diseases, Sapienza University of Rome, 00185 Rome, Italy; lorenza.lia@uniroma1.it (L.L.); eleonora.ricci@uniroma1.it (E.R.); corrado.colaprico@uniroma1.it (C.C.); eleonora.dilegge91@gmail.com (E.D.L.); augusto.faticoni@uniroma1.it (A.F.); 2Department of Experimental Medicine-Medical Pathophysiology, Food Science and Endocrinology Section, Sapienza University of Rome, 00185 Rome, Italy; lorenzomaria.donini@uniroma1.it

**Keywords:** COVID-19, lockdown, quarantine, nutrition, BMI, lipid, workers, hospital

## Abstract

Background: on the 9 March 2020, the Italian government declared a state of lockdown on the entire national territory aimed at reducing the spread of SARS-CoV-2, causing strong repercussions for people’s lifestyles. The aim of the study was to analyze the impact of the lockdown on the nutritional status and lipid profile of employees of an Italian teaching hospital. Methods: an observational retrospective cohort study was carried out at the Department of Occupational Medicine of the Umberto I General Hospital of Rome, including all employees who underwent two consecutive occupational medical examinations before and after the first lockdown (9 March 2020–18 May 2020). Employee medical records were used as a data source. Results: 1014 employees were involved in the study (50.6% nurses, 31% physicians, 14.8% technical staff, 3.6% administrative staff). Post lockdown BMI, total cholesterol and LDL values increased statistically significantly compared to pre lockdown ones. Nurses showed a significant association with increased BMI (*p* < 0.001), while workers with heart disease were inversely associated with total cholesterol (*p* < 0.001) and LDL (*p* < 0.001). Conclusion: this study showed that lockdown had a significant impact on employees’ lifestyles. Further studies are needed to understand changes in health-related behaviors, such as diet and physical activity, of specific categories of workers over time under lockdown conditions.

## 1. Introduction

On 11 February, the WHO declared that the respiratory disease caused by the new coronavirus was called COVID-19 (Corona Virus Disease) [1] and on the 11 March, it announced that the outbreak could be characterized as a pandemic [2]. 

In Italy, on the 9 March 2020, the government declared a state of lockdown for the entire national territory [3]. During this period of confinement, the population was allowed to move only for necessary reasons (work, purchase of food, health reasons and basic necessities) with a view to social distancing. The closure of all commercial activities not deemed necessary, museum closures, cultural sites and sports center closures, the interruption of sports activities and public events were imposed. The most restrictive confinement measures in Italy lasted until the 18 May 2020 (phase 2), but this unprecedented situation disrupted people’s daily routines and lifestyles [4]. Due to these restrictions, people found it difficult to follow a healthy and balanced diet, preferring the consumption of high-calorie convenience foods, snacks and junk food, instead of fresh food, such as fruits and vegetables [5]. An Italian study revealed that, during the lockdown period, 46.1% of people increased their consumption of “comfort” food, rich in sugars and fats, while 19.5% recorded an increase in their body weight [6]. Researchers from the Council for Agricultural Research and Economics (CREA) analyzed some studies conducted in Italy, Portugal, Spain, France and Poland, based on the formulation and dissemination of different types of questionnaires, with the aim of comparing the data obtained and highlighting the criticality and positive aspects that emerged in this particular period in reference to eating habits and physical exercise [7]. The analysis took into consideration various lockdown periods in 2020. The general results show that the quarantine had effects on both eating habits and physical activity, highlighting an increase in food consumption and a reduction in physical activities, resulting in weight gain. This situation is undoubtedly linked to the many hours spent at home and the need for comfort food to cope with the anxiety caused by the exceptional situation. Furthermore, ease of access to food during home confinement and altered emotional states may have caused an increase in bingeing episodes in people with obesity [8]. Among HCWs, studies conducted in a teaching hospital in Rome demonstrated the time spent on physical exercise in this category of workers was shorter or much shorter in 88.3% of people in the first, 81.8% in the second and 79.3% in the third waves [9,10,11].

While for some people it was possible to work from home, others, such as healthcare workers, had to continue working regularly in person to deal with the health emergency. If the work of this professional category during the quarantine period maintained a rather high pace on average, all or almost all the refreshment services they used to use during the work shift or during breaks were suspended if unable to maintain safety standards [3]. Moreover, food discomfort was not limited only to working hours; the difficulty in finding food leads to facing long fasts or eating high-calorie take-away foods, but it also continues once back home.

There are currently many studies investigating the impact of restrictions due to the COVID-19 pandemic on lifestyle changes in the general population, but changes in the lifestyle behaviors of a specific population with an expected knowledge of nutrition, health and lifestyle management, such as healthcare professionals, including doctors, nurses and other healthcare providers, is an area still under investigation. 

This study aims to analyze the impact of the lockdown imposed during the first wave of COVID-19 pandemic on the nutritional status and lipid profile of Italian teaching hospital employees, by comparing the data collected during medical surveillance visits in the pre and post lockdown periods.

## 2. Materials and Methods

### 2.1. Study Design and Setting

An observational retrospective cohort study was carried out. The Strengthening the Reporting of Observational Studies in Epidemiology (STROBE) was applied to perform the research [12].

This study took place at the Department of Occupational Medicine of the Umberto I General Hospital in Rome and included all hospital employees who underwent two consecutive occupational medical examinations, one before and one after the first lockdown established in Italy (9 March 2020–18 May 2020).

Specifically, patients who underwent a medical examination from May 2018 to March 2020 and a subsequent medical examination from May 2020 to May 2021 were considered eligible to be included in the study. Employee medical records were used as a data source. Medical records of workers who did not meet the above criteria were excluded.

### 2.2. Study Outcomes and Variables

The following variables were collected for each patient included in the study: sex, age, date of pre-lockdown medical examination, date of post-lockdown medical examination, professional role (physicians, nurses, technical staff, administrative staff), judgment of suitability for the job (with or without limitations), exposure to biological risk, exposure to chemical risk, visual display unit (VDU) operators, manual handling of loads (MHL)/manual patient handling (MPH), night shifts working, smoking status, alcohol consumption (AUDIT-C questionnaire score [13]), physical activity, chronic diseases (diabetes, hypertension, ischemic heart disease, thyroid disease, oncological pathologies) and/or other pathologies (such as COVID-19).

Moreover, the following parameters were considered for both medical visits: height and weight (measured during both medical visits) used to calculate BMI [weight (kg) ÷ height^2^ (m)], total cholesterol (mmol/L), HDL cholesterol (mmol/L), triglycerides (mmol/L) and LDL cholesterol calculated through Friedewald’s formula [total cholesterol—(HDL + TR/5)] [14].

The anonymity of the workers was guaranteed for the creation of the database.

### 2.3. Statistical Analysis 

All analyses were performed using SPSS for Windows (Statistical Package for the Social Sciences, Version 27; SPSS, Inc., Chicago, IL, USA). A descriptive analysis was conducted using medians, and minimum and maximum values for quantitative variables. For categorical variables, absolute frequencies and percentages were computed. BMI, total cholesterol and LDL values before and after lockdown were compared using Wilcoxon’s signed-rank test.

The difference between the pre and post lockdown variables (BMI, total Cholesterol, LDL cholesterol) was calculated to obtain the delta (Δ) variable (ΔBMI = BMI post-BMI pre, ΔTCho = Total Cholesterol post-Total Cholesterol pre, ΔLDL = LDL post-LDL pre).

The nonparametric Mann–Whitney U test was applied for two-group comparisons, and the Kruskal–Wallis test was used for comparisons of more than two groups.

The Spearman’s rank correlation coefficient was computed to estimate the direct or indirect relation between the ranks of the following variables: age, alcohol consumption, BMI Pre, BMI Post, Total Cholesterol Pre, Total Cholesterol Post, LDL Pre, LDL Post.

Multivariate linear regression models were developed using Stepwise with backward elimination of non-significant variables (probability to entry *p*  < 0.05) with ΔBMI, ΔTCho, ΔLDL as dependent variables. The results are presented using beta coefficients (*p*-values).

Three linear regression models were created in order to study the relationships between the dependent (ΔBMI, ΔTCho, ΔLDL) and independent variables (sex, age, professional role, judgment of suitability for the job, exposure to biological risk and chemical risk, VDU, MHL/MPH, night shifts working, smoking status, alcohol consumption, diabetes, hypertension, ischemic heart disease, thyroid disease, oncological pathologies). The same models were performed, stratifying the analysis by gender and age (age groups < 52 years and ≥52 years) to assess possible effect modification. 

Moreover, a further linear regression model was developed only for a subgroup (46.5% of the sample) by adding a physical activity to the independent variables. 

The goodness of fit of the different linear regression models performed was evaluated using the R^2^ statistic. The significance threshold was set at *p* < 0.05 for all analysis.

## 3. Results

A total of 1014 employees were involved in the study, 617 of which (60.8%) were women, with a median age of 52.35 (21.30–67.49). The majority of the sample consisted of 513 (50.6%) nurses, including other categories belonging to HCW such as physiotherapists and midwives, followed by 397 (31%) physicians, 150 (14.8%) technical staff (laboratory technicians, biologists, pharmacists) and 37 (3.6%) administrative staff. The medians of BMI, total cholesterol and LDL post lockdown slightly increased compared to pre lockdown values, as shown in Table 1.

### 3.1. Univariate Analysis

Univariate analysis of ΔBMI obtained significant differences only between professional roles (*p* = 0.001), MHL/MPH operators (*p* = 0.002) and smoking status (*p* = 0.030). 

The Mann–Whitney test detected statistically significant differences across the presence or absence of heart disease in Δ total cholesterol (*p* = 0.015) and ΔLDL (*p* = 0.035), showing that medians are significantly reduced in workers who have suffered from ischemic heart disease. Comparison between groups of the other variables included in the analysis did not report significant results for both Δ Total Cholesterol and ΔLDL. The results of the univariate analysis are shown in Table 2.

### 3.2. Bivariate Analysis

Correlation analysis, illustrated in Table 3, showed a statistically significant association between all variables, except for AUDIT-C score. No significant association was found between total cholesterol post lockdown and BMI pre lockdown (Spearman’s rho = 0.01; *p* = 0.55). 

### 3.3. Multivariate Analyses

#### 3.3.1. Multiple Linear Regression Analysis of ΔBMI

Multiple linear regression analysis with ΔBMI as a dependent variable, represented in Appendix A, showed a significant direct association with nurses (β = 0.119; *p* ≤ 0.001) and workers with ischemic heart disease (β = 39.3; *p* = 0.020). The linear regression model developed for the subgroup with physical activity showed a direct association between ΔBMI and nurses (β = 0.166; *p* = 0.001), confirming the relationship resulting in the model with the whole study population, and a significant negative association with AUDIT-C score (β = −0.114; *p* = 0.015). In the stratified analysis, gender and age resulted as effect modifiers (Appendix A).

#### 3.3.2. Multiple Linear Regression Analysis of ΔLDL

As can be seen in Appendix A, physicians had a significant relationship with ΔLDL (β = 0.078; *p* = 0.013), unlike employees exposed to chemical risk (β = −0.062; *p* = 0.047) and those with heart disease (β = −0.123; *p* ≤ 0.001) who showed an inverse association. In the PA subgroup analysis, physicians were directly associated with ΔLDL (β = 0.180; *p* = 0.001), while the variables concerning chronic diseases and physical activity were inversely related, indicating that more physically active workers have a lower LDL value. Moreover, this is the model with a better performance (R^2^ = 0.107) compared to the other linear regression models created in this study (Appendix A). In stratified analysis, gender and age resulted as effect modifiers (Appendix A).

#### 3.3.3. Multiple Linear Regression Analysis of Δ Total Cholesterol

In linear regression analysis with Δ total cholesterol as a dependent variable, represented in Appendix A, the only variable with a statistically significant association was heart disease (β = −0.133; *p* ≤ 0.001). In the PA subgroup analysis, heart disease reported the same association (β = −0.148; *p* = 0.001), along with oncological pathologies (β = −0.194; *p* ≤ 0.001) and physical activity (β = −0.097; *p* = 0.033). Gender and age were also effect modifiers for Δ total cholesterol (Appendix A).

## 4. Discussion

This study was conducted to investigate changes due to the lockdown imposed during spring 2020 by the Italian government to limit the spread of COVID-19 in BMI and lipid profile values. The population examined consisted of 1014 employees of the Umberto I General Hospital and included healthcare workers, who fell into the essential categories not subjected to home confinement, but also administrative staff. The main results demonstrated that there was a significant increase in BMI, total cholesterol and LDL values after lockdown compared to pre lockdown. These findings are in agreement with those of previous studies conducted in different populations. It has been widely highlighted that the security measures adopted to limit the spread of the pandemic, such as lockdowns and home quarantine, have caused many changes in lifestyles [6] and conditioned personal habits, including changes in behaviors and food choices [5], the limitation of the possibilities to carry out physical activities and the increase of alcohol consumption [15]. A Chinese study, carried out on a sample of health workers from Hubei province during the first wave of pandemic, found that 26.2% of the participants reported weight gain, while 22.9% reported weight loss. The authors further stated that a large proportion of healthcare workers had unbalanced diets [16]. A survey carried out with Brazilian Urologists showed modifications in health and lifestyle. In particular, one third of the participants reported weight gain (32.9%) and more than half of them reported reduced physical activity (60.0%) [17]. Another study carried out in Brazil among cardiologists found that 44% of participants gained weight, and among these 13% gained more than 3 kg [18]. Another study conducted in Southern Italy before (until December 2019) and during (until May 2020) the first wave of the pandemics, on 291 Italian nurses, also demonstrated weight gain [19].

Of all professionals involved in the current study, nurses were the only ones to have shown a statistically significant association with ΔBMI, along with workers affected by ischemic heart disease, indicating that their condition is associated to an increase in post-lockdown BMI values compared to pre-lockdown ones. This can be explained both because nursing staff was the most represented employee category in the sample and because they were probably among the most stressed workers in the management of the COVID-19 pandemic. 

It is well known that frontline health workers are among the groups most exposed to the risk of mental health problems. Symptoms of emotional and intense psychological distress, anxiety, depression, nervousness, irritability, persistent insomnia and symptoms referable to post-traumatic stress disorder are common, along with painful feelings of guilt and sadness and fear and worry about infecting themselves and their families [20,21,22,23]. A longitudinal study carried out at Gemelli Teaching hospital in Rome between spring 2020 and spring 2021 among intensive care physicians showed a lack of time for physical activity and meditation, and compassion fatigue, as well as increased workload, isolation at work and in their social life. Moreover, in this cohort of physicians, stress was inversely associated with the perception of justice in safety procedures and directly correlated with work isolation. Finally, occupational stress was found to be significantly associated with anxiety, depression and burnout [11].

Although there are not mental health indicators, during the lockdown there was an increase in BMI. This could be associated with three factors: staying at home means eating more, not performing physical activity, and being stressed, understood as a state of mental health. All of that can have an influence on bad habits. For these reasons, we can say that the relationship between mental state and nutritional state is an issue. The results of a Chinese study carried out during the COVID-19 epidemic involving 1257 healthcare workers who assisted patients in COVID-19 wards and in wards placed in the second and third line, reported important percentages of depression, anxiety, insomnia and distress, with particular severity especially for nurses and women [24]. Mood disturbances and emotional changes affect the choice and quantity of food taken, directed towards an unhealthy and unbalanced diet [25]. In moments of greatest emotional stress, many seek fulfillment and relief in their favorite food. The choice normally falls on foods rich in calories in the form of sugars and fats such as biscuits, chips, pizza, ice cream and chocolate, better known as “comfort food”, because they are tasty and seem to give a feeling of immediate pleasure and satisfaction [26,27]. Recent evidence has found a significant shift in eating behaviors towards unhealthy overeating, associated with weight gain, among people who have felt more stressed or depressed during COVID-19 confinement [28,29]. 

Regarding lipid profile, unlike BMI, workers suffering from ischemic heart disease showed a significant association with the reduction in both total cholesterol and LDL cholesterol values. The same type of association was also found in the analyses stratified by gender and age among men and those aged over 52. However, it should be taken into consideration that cardiopathic patients generally undergo therapy for the control of blood cholesterol levels, such as statins or other lipid-lowering drugs, so the results obtained could be influenced by this factor. Nevertheless, an Italian study reported a significant increase during lockdown in total cholesterol and LDL in patients at high cardiovascular risk, who had been prescribed drug therapy with statins, a rigorous physical activity program and a personalized Mediterranean diet model. The authors stated that all patients discontinued the prescribed physical activity program, and the Mediterranean Diet alone failed to compensate and maintain healthy lipid profiles [30]. 

The professional role of medical doctors, on the other hand, resulted in being associated with an increase in total cholesterol following the lockdown of the first wave of COVID-19, compared to other professionals who have not shown significant associations.

The further analyses carried out in the subgroup of the sample for which it was possible to collect data on physical activity (472 out of 1014 employees included, 46.5% of the total), largely confirmed the associations obtained in the analyses on the total sample. In this subgroup, all the chronic diseases considered in this study (diabetes, ischemic heart disease, thyroid disease, oncological pathologies) have been shown to significantly influence ΔLDL values. Workers who suffer from these pathologies seem to be related to a reduction in LDL levels compared to their healthy counterparts. These data are not in agreement with those observed in a study in which diabetic subjects had an increase in LDL cholesterol along with glucose and triglycerides, compared to non-diabetics [31]. In our case, it can be assumed that people affected by these diseases were afraid of becoming seriously ill with COVID-19 and have improved their eating habits.

Moreover, from our study we found that physical activity was inversely associated with all three outcome variables (ΔBMI, ΔTCho, ΔLDL). Employees who managed to remain physically active during the lockdown, despite limitations and restrictions, were associated with a decrease in BMI and total cholesterol and LDL levels after the lockdown. This confirms the fundamental role of physical activity in preventing weight gain and a worsening of the lipid profile. A previous observational study conducted on a population of healthy adults evaluating the effect of COVID-19 lockdown on changes in eating habits, physical activity and serum markers, highlighted a worsening in diet quality and physical activity levels during lockdown, and an increase in serum glucose, total cholesterol and LDL post lockdown. However, no changes in body composition were reported, perhaps due to a decrease in energy intake during the lockdown [32]. 

The data emerging from the stratified analysis by gender and age relating to night shifts and AUDIT-C deserve a separate consideration. Shift workers were demonstrated to be associated with lower ΔBMI (among men), Δ total cholesterol and ΔLDL (among the younger age group). Shift work and night work are known to be associated with weight gain, as reported by several studies, many of them conducted on a population of healthcare workers [33,34,35]. Nevertheless, a cross-sectional study conducted by Know et al. found male workers on night shifts were associated with weight loss. The authors motivated these findings by attributing weight loss to depressive symptoms, gastrointestinal disorders and loss of appetite due to stress, common in shift workers [36]. These explanations could also be easily applicable to the workers in our study, under the burden of managing the pandemic. 

AUDIT-C reported the same relationship with ΔBMI, among women and in the subgroup with physical activity, indicating that a higher AUDIT-C score, and therefore more alcohol consumption, was associated with a decrease in BMI post lockdown. Alcohol provides a large amount of calories that are added to the calories supplied by food and can therefore contribute to weight gain. Furthermore, the calories consumed in the form of alcohol are defined as “empty”, since each calorie consumed is not associated with any nutrient useful for the body. Ethanol is a non-essential substance for the body as it is non-nutritious and has a high energy content. However, there appears to be an “alcohol paradox”. In fact, it seems from some studies that the intake of alcohol in addition to a normal calorie diet does not lead to weight gain [37,38]. In addition, the amount of alcohol consumed should also be considered: light to moderate alcohol intake may be more likely not to lead to weight gain, while excessive and regular alcohol consumption appears to be more related to weight gain [39,40]. It is worth noting that most of the workers included in our study had a low-risk AUDIT-C score between 0 and 2, while only a very small percentage (2.7%) had a score between 4 and 5 associated with possible risky alcohol consumption.

### Strengths and Limitations

Several observational studies conducted through surveys in various European and non-European countries have shown changes in the weight and nutritional status of the population during COVID-19 home confinement. Nevertheless, this is the first cohort study carried out on a population of workers that includes healthcare professionals with the aim of evaluating the impact of COVID-19 lockdown on nutritional status and lipid profile. Previous studies conducted on this population group have focused mainly on the effects on mental health due to the management of the COVID-19 pandemic and related psychological and psychosocial support actions.

However, this study has some limitations that need to be acknowledged. First of all, it was not possible to obtainimportant data such as physical activity for all employees included, as it is not reported in all medical records. For the same reason, another limitation of this study is related to the lack of data on the participants’ eating habits before and after the lockdown period imposed by the Italian government in 2020. Furthermore, we were unable to assess workers’ perceived stress levels, and how they impacted their lifestyles. 

Despite this, the data collected come from medical examinations and clinical analyses and therefore can be considered reliable and accurate.

## 5. Conclusions

The data from this study showed that participants’ BMI and lipid profile changed, highlighting that lockdown had a significant impact on their lifestyles.

Nutrition plays a fundamental role in people’s health, especially in those exposed to a high intensity of work, such as health workers in this period of pandemic emergency. Further studies are needed to evaluate changes in the eating habits of specific categories of workers over time under lockdown conditions.

A thorough understanding of changes to health-related behaviors, such as diet and physical activity, in this context would provide important information for designing targeted health promotion actions and advice tailored to the population.

## Figures and Tables

**Table 1 ijerph-19-04549-t001:** Characteristics of the sample.

Variables	N (%); Median (Min-Max)
**Age**	52.35 (21.30–67.49)
**Gender**	
Male	397 (39.2%)
Female	617 (60.8%)
**Role**	
Physicians	314 (31.0%)
Nurses	513 (50.6%)
Technicians	150 (14.8%)
Administratives	37 (3.6%)
**BMI**	
Pre	24.11 (15.82–50.00)
Post	24.34 (16.00–48.93)
*p*	*
**LDL**	
Pre	3.32 (1.03–6.47)
Post	3.42 (−0.76–7.39)
*p*	*
**Cholesterol**	
Pre	5.15 (2.43–9.08)
Post	5.32 (0.82–9.75)
*p*	*

* *p* ≤ 0.001.

**Table 2 ijerph-19-04549-t002:** Results of the univariate analysis.

Variables	∆ BMI		∆ Total Cholesterol		∆ LDL	
	Median (Min-Max)	*p*	Median (Min-Max)	*p*	Median (Min-Max)	*p*
**Gender**		0.152		0.999		0.935
Male	0.00 (−6.04–8.59)		0.16 (−5.02–3.34)		0.12 (−4.60–3.09)	
Female	0.16 (−5.86–7.84)		0.14 (−5.00–2.94)		0.11 (−5.00–2.80)	
**Role**		*		0.292		0.066
Physicians	0.00 (−5.62–6.00)		0.23 (−2.46–3.10)		0.17 (−3.24–2.86)	
Nurses	0.34 (−6.04–8.59)		0.12 (−5.02–3.34)		0.09 (−5.00–3.09)	
Technicians	0.00 (−4.69–6.81)		0.09 (−3.41–2.22)		0.01 (−3.27–2.04)	
Administratives	0.00 (−3.46–4.41)		0.14 (−2.00–1.58)		0.19 (−2.20–1.55)	
**Judgment**		0.692		0.992		0.759
Eligible	0.00 (−6.04–8.59)		0.14 (−5.02–3.34)		0.11 (−4.60–3.09)	
With Limitation	0.10 (−5.86–7.81)		0.10 (−5.00–3.10)		0.11 (−5.00–2.86)	
**Biological Risk**		0.224	0.12 (−0.18–1.08)	0.649		0.864
No	0.69 (−0.38–5.47)		0.15 (−5.02–3.34)		0.20 (−0.54–0.48)	
Yes	0.00 (−6.04–8.59)				0.11 (−5.00–3.09)	
**VDU**		0.422		0.199		0.166
No	0.00 (−6.04–8.59)		0.17 (−5.02–3.34)		0.12 (−5.00–3.09)	
Yes	0.30 (−4.69–6.81)		0.08 (−2.86–2.94)		0.03 (−2.66–2.80)	
**MHL/MPH**		**		0.845		0.708
No	0.00 (−5.86–8.59)		0.16 (−5.00–3.10)		0.12 (−5.00–2.86)	
Yes	0.34 (−6.04–7.81)		0.14 (−5.02–3.34)		0.11 (−4.60–3.09)	
**Night work**		0.616		0.468		0.667
No	0.00 (−5.86–8.59)		0.17 (−5.00–3.10)		0.12 (−5.00–2.86)	
Yes	0.00 (−6.04–7.44)		0.14 (−5.02–3.34)		0.10 (−4.60–3.09)	
**Chemical Risk**		0.123		0.152		0.101
No	0.00 (−6.04–8.59)		0.16 (−5.02–3.34)		0.12 (−5.00–3.09)	
Yes	0.00 (−2.42–5.47)		0.07 (−2.40–2.02)		−0.01 (−3.24–2.04)	
**Smoker**		**		0.841		0.504
No	0.00 (−5.62–8.59)		0.14 (−5.00–2.94)		0.10 (−5.00–2.80)	
Yes	0.31 (−6.04–7.81)		0.19 (−5.02–3.34)		0.14 (−4.60–3.09)	
**Diabetes**		0.865		0.19		0.140
No	0.00 (−6.04–8.59)		0.16 (−5.02–3.34)		0.12 (−5.00–3.09)	
Yes	0.32 (−3.46–3.91)		0.08 (−2.86–1.84)		−0.10 (−4.16–1.83)	
**Hypertension**		0.431		0.606		0.3
No	0.00 (−5.86–8.59)		0.14 (−5.02–2.41)		0.10 (−4.60–2.09)	
Yes	0.00 (−6.04–4.84)		0.18 (−5.00–3.34)		0.15 (−5.00–3.09)	
**Heart Disease**		0.146		**		**
No	0.00 (−6.04–8.59)		0.16 (−5.02–3.34)		0.12 (−5.00–3.09)	
Yes	0.35 (−0.67–4.24)		−0.42 (−3.41–0.59)		−0.28 (−3.27–0.72)	
**ThyroidDisease**		0.134		0.104		0.099
No	0.00 (−5.86–8.59)		0.17 (−5.02–3.34)		0.12 (−5.00–3.09)	
Yes	0.30 (−6.04–5.47)		0.06 (−2.94–2.82)		0.02 (−4.16–2.49)	
**Cancer**		0.59		0.31		0.224
No	0.00 (−6.04–8.59)		0.16 (−5.00–3.34)		0.12 (−5.00–3.09)	
Yes	0.00 (−5.86–3.38)		0.04 (−5.02–2.16)		−0.02 (−4.60–2.09)	

* *p* ≤ 0.001; ** *p* ≤ 0.05

**Table 3 ijerph-19-04549-t003:** Results of the bivariate analysis.

Variables	Age	AUDIT-C			BMI Pre		BMI Post		LDL Pre		LDL Post		TCho Pre		TCho Post	
	Corr. Coeff.	*p*	CorrCoeff.	*p*	CorrCoeff.	*p*	CorrCoeff.	*p*	CorrCoeff.	*p*	CorrCoeff.	*p*	CorrCoeff.	*p*	CorrCoeff.	*p*
**TCho Post**	0.25	*	0.01	0.68	0.01	0.55	0.06	**	0.65	*	0.9	*	0.7	*	-	
**TCho Pre**	0.24	*	0	0.89	0.06	**	0.08	*	0.88	*	0.62	*	-		0.7	*
**LDL Post**	0.25	*	0.01	0.7	0.14	*	0.18	*	0.71	*	-		0.62	*	0.9	*
**LDL Pre**	0.26	*	0.01	0.74	0.18	*	0.2	*	-		0.71	*	0.88	*	0.65	*
**BMI Post**	0.16	*	0.01	0.53	0.92	*	-		0.2	*	0.18	*	0.08	*	0.06	**
**BMI Pre**	0.2	*	0.03	0.27	-		0.92	*	0.18	*	0.14	*	0.06	**	0.01	0.55
**AUDIT-C**	0.04	0.14	-		0.03	0.27	0.01	0.53	0.01	0.74	0.01	0.7	0	0.89	0.01	0.68
**Age**	-	0.04		0.14	0.2	*	0.16	*	0.26	*	0.25	*	0.24	*	0.25	*

* *p* ≤ 0.001; ** *p* ≤ 0.05.

## Data Availability

Data are available upon request.

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
