# Peer review of "Assessment of the Impact of COVID-19 Lockdown on the Nutritional Status and Lipid Profile of Employees in a Teaching Hospital in Rome: A Retrospective Cohort Study"

_ijerph, 2022, doi:10.3390/ijerph19084549_

Round 1

Reviewer 1 Report

To the authors.

It is important work to use all available data, so it was good to see that presented here. It is also important to have "proof" to go along with the assumptions that occurred throughout the pandemic. The statistical methods seemed solid and robust for this type of data analysis. As there was not a food frequency or any type of dietary data collected, it is extremely important that the authors do not make misleading or inappropriate conclusions related to this. Despite seeing an increased BMI, without food records it is not possible to attribute this to certain components of the diet. I would also caution the authors when including components related to mental health, as there were not any outcomes or tools or measured used in this study to measure mental health. 

Specific and Overall Comments

  • Depending on the requirements of the journal, some decimal formatting may need to be adjusted (1014 employees --> 1,014 employees; 50,6% nurses --> 50.6% nurses).
  • Due to restrictions in activities, stress, etc. it is not surprising that there are effects to food consumption and body wait. I am left with wondering "so what"? If restrictions are needed again, what can we add or change that might improve the health of those living through the restrictions? 
  • The findings related to heart disease are interesting, and potentially quite significant. I have a feeling these disease states also worsened for other conditions. Was it possible to adjust for time in these analyses? For example, if 1-2 years of time has passed, it is expected that without treatment, a condition would get worse. So, is the worsening due to the pandemic and restrictions, or more simply related to the passing of time and worsening of the natural condition? 
  • Line 22-23 - grammar to be adjusted
  • Unclear what "chemical" means in Table 2?
  • I found the text from lines 204-224 quite difficult to read with the interruption of the statistics. Is there a different way to present this data, or make reference to the tables? It was difficult to get a good sense of the results with too many things to keep in mind. Are there some results worth highlighting, while the rest are ok to keep listed in the table? 
  • Line 250-254. Although this point is very relevant to the pandemic in general, I do not think it is relevant to the article. No other mental health indicators are measured or mentioned, so it feels out of place. 
  • Lines 265-269 could be compressed into one sentence. 
  • Line 296- was this results from the study, or in reference to another study? This was not clear. 
  • Line 353 - since a food frequency questionnaire was not collected, it is not possible to associate the change in weight with higher calorie foods, or even specific types of foods. This is not a conclusion possible from this work, and should be removed. 

Author Response

Specific and Overall Comments

  • Depending on the requirements of the journal, some decimal formatting may need to be adjusted (1014 employees --> 1,014 employees; 50,6% nurses --> 50.6% nurses).

Answer: Thanks to the Reviewer for these indications, we have done the corrections directly on the article

  • Due to restrictions in activities, stress, etc. it is not surprising that there are effects to food consumption and body wait. I am left with wondering "so what"? If restrictions are needed again, what can we add or change that might improve the health of those living through the restrictions? 

Answer: Thanks to the Reviewer for this question. Our study is one of the first about this topic. As we wrote in the conclusion of the article, it is necessary to give differents sources and data analysis to the governments and other relevant decision makers who are aware of the health effects of not only SARS-CoV-2 infection, but also of the restrictions imposed to limit its spread, so they can plan policies to minimize its adverse effect by implementing effective dietary and physical activity interventions. Nutritional education interventions can lead to important changes in behaviour and nutritional choices and help develop a diet that promotes health, following the dictates of the Mediterranean Diet

  • The findings related to heart disease are interesting, and potentially quite significant. I have a feeling these disease states also worsened for other conditions. Was it possible to adjust for time in these analyses? For example, if 1-2 years of time has passed, it is expected that without treatment, a condition would get worse. So, is the worsening due to the pandemic and restrictions, or more simply related to the passing of time and worsening of the natural condition? 

Answer. Thanks to the Reviewer for this question. Healthcare workers are naturally exposed to work pressures and stress, which have increased with the lockdown. From the analyzes conducted there is a strong correlation with the worsening of cardiovascular diseases: this leads us to think that it is not only related to the natural worsening dictated by age. In addition cardiovascular risk is an epiphenomenon and not the focus of the article.

  • Line 22-23 - grammar to be adjusted

Answer: As reported by the Reviewer we have modified the field.

  • Unclear what "chemical" means in Table 2?

Answer: we agree with the reviewer. "Chemical" stands for chemical risk exposure. To make it more understandable, in all tables we have added "risk" in the "biological" and "chemical" variables.

  • I found the text from lines 204-224 quite difficult to read with the interruption of the statistics. Is there a different way to present this data, or make reference to the tables? It was difficult to get a good sense of the results with too many things to keep in mind. Are there some results worth highlighting, while the rest are ok to keep listed in the table? 

Answer: we agree with the reviewer on the need to make it easier to read the results. We have removed the stratified analysis paragraph and inserted what we believe is important to highlight within the paragraphs relating to multivariate analysis in general, together with the references to the tables.

  • Line 250-254. Although this point is very relevant to the pandemic in general, I do not think it is relevant to the article. No other mental health indicators are measured or mentioned, so it feels out of place. 

Answer: We have explained our point of view in the discussion as reported in the article,

  • Lines 265-269 could be compressed into one sentence.

Answer: we have modified the period.

  • Line 296- was this results from the study, or in reference to another study? This was not clear. 

Answer: we have modified the sentences

  • Line 353 - since a food frequency questionnaire was not collected, it is not possible to associate the change in weight with higher calorie foods, or even specific types of foods. This is not a conclusion possible from this work, and should be removed.

Answer: Thanks to the Reviewer, we have removed the sentences

Reviewer 2 Report

Assessment of the impact of COVID-19 lockdown on the nutritional status and lipid profile of employees in a teaching hospital in Rome: a retrospective cohort study analyzes the impact of the lockdown on the nutritional status and lipid profile of the employees of an Italian teaching hospital.

I would recommend considering the following suggestions:

The introduction of the manuscript is too long. I would recommend shortening it by removing the initial part on the onset of the covid and going straight to the point: the measures taken by the Italian government to prevent the spread.

I would also suggest shortening of the text relatively to the research that was carried out worldwide. On the contrary I think it is necessary to explain much better why there is a lack of such a study specifically for health professionals and, most of all, why is important to study that!

Relatively to possible reasons of weight gain, not only the canteens were closed, you should also add something about the relationship psychological state / stress and hunger/eating as well as the fact that often not only the health sector workers were not able to work from home but they even work many extra hours.

lines 110-116 simply list the data collected and the measurements made. The comparisons were made retrospectively, in 2018 no one knew about the Covid, so the data were not collected to compare pre and post pandemic.

The presentation of the results must be shortened, streamlined, and made more understandable. The results should be presented either in the table or in the text, not both.

Statistical analysis is very detailed, perhaps not all presented results presented are necessary. I would suggest to resume the most important ones.

In the tables I suggest to use the symbols to indicate only statistically significant differences (distinguishing between <0.01 and <0.05) and then to specify in the note below the table the meaning of the symbols ( for example * = 0.01; ** = 0.05).

The results of the multivariate analysis: I suggest to summarize them in a single paragraph and not to show them in the sub-paragraphs.

I recommend reviewing the discussion in the light of the suggested changes.

English must be checked.

Author Response

The introduction of the manuscript is too long. I would recommend shortening it by removing the initial part on the onset of the covid and going straight to the point: the measures taken by the Italian government to prevent the spread.

Answer: As repquested by the Reviewer we have modified the introduction.

I would also suggest shortening of the text relatively to the research that was carried out worldwide. On the contrary I think it is necessary to explain much better why there is a lack of such a study specifically for health professionals and, most of all, why is important to study that!

Answer:  Thanks to the Reviewer for the question. The reason why there is a lack of studies with this focus is that it is difficult to follow-up people (in this case healthcare workers) in cohorts to be able to carry out epidemiological studies. At the teaching hospital Policlinico Umberto 1, we were able to recruit a large number of health professionals thanks to the surveillance of occupational medicine. In addition, the strength of our work is that we use the medical records and not only a questionnaire, naturally exposed to bias.

Relatively to possible reasons of weight gain, not only the canteens were closed, you should also add something about the relationship psychological state / stress and hunger/eating as well as the fact that often not only the health sector workers were not able to work from home but they even work many extra hours.

Answer: Thanks to the Reviewer for this focus. During the Lockdown there was an increase in BMI. This is due to three factors: staying at home means eating more, do not do physical activity, and be stressed, understood as a state of mental health.  All of  that influences bad habits. For this  reason we can say that there is a correlation between mental state and nutritional state.

lines 110-116 simply list the data collected and the measurements made. The comparisons were made retrospectively, in 2018 no one knew about the Covid, so the data were not collected to compare pre and post pandemic.

Answer: the data listed in the paragraph indicated by the reviewer are parameters that are usually measured and entered in the medical records of patients during occupational medical examinations. For the purposes of our study, the same parameters (both those relating to the pre lockdown visit and those relating to the post lockdown visit) were collected from the medical records of all participants and entered in the database for the analysis of interest.

To avoid misunderstandings, the sentence "in order to allow an evaluation of the differences pre and post lockdown" (line 110) has been removed. We thank the reviewer for pointing this out to us.

The presentation of the results must be shortened, streamlined, and made more understandable. The results should be presented either in the table or in the text, not both.

Answer: we agree with the reviewer on the need to simplify the presentation of the results. We have removed some results that we felt were not essential but can be consulted in the tables.

Statistical analysis is very detailed, perhaps not all presented results presented are necessary. I would suggest to resume the most important ones.

Answer: we thank the reviewer for the comment on our analysis. As reported above, we have kept in the text only the results that we consider most important and eliminated those that we believe can be consulted in the tables

In the tables I suggest to use the symbols to indicate only statistically significant differences (distinguishing between <0.01 and <0.05) and then to specify in the note below the table the meaning of the symbols ( for example * = 0.01; ** = 0.05).

Answer: we thank the reviewer for the suggestion.  We have replaced the significant p values with the symbols * for p values ≤0.001 and ** for p values ≤0.05.

The results of the multivariate analysis: I suggest to summarize them in a single paragraph and not to show them in the sub-paragraphs.

Answer: thanks for the suggestion. We have removed the stratified analysis paragraph and inserted what we believe is important to highlight within the paragraphs relating to multivariate analysis in general, together with the references to the tables. However, it seems more appropriate to keep the paragraphing of the multivariate analysis relating to the three dependent variables for greater clarity and understanding of the results.

I recommend reviewing the discussion in the light of the suggested changes.

English must be checked.

Answer: done as requested
